# Contraction of basal filopodia controls periodic feather branching via Notch and FGF signaling

Dongyang Cheng[1], Xiaoli Yan[1], Guofu Qiu[1], Juan Zhang[1], Hanwei Wang[1], Tingting Feng[1], Yarong Tian[1], Haiping Xu[2], Meiqing Wang[2], Wanzhong He [3], Ping Wu[4], Randall B Widelitz [4], Cheng-Ming Chuong[4] & Zhicao Yue [1]

Branching morphogenesis is a general mechanism that increases the surface area of an organ. In chicken feathers, the flat epithelial sheath at the base of the follicle is transformed into periodic branches. How exactly the keratinocytes are organized into this pattern remains unclear. Here we show that in the feather follicle, the pre-branch basal keratinocytes have extensive filopodia, which contract and smooth out after branching. Manipulating the filopodia via small GTPases *RhoA/Cdc42* also regulates branch formation. These basal filopodia help interpret the proximal-distal FGF gradient in the follicle. Furthermore, the topological arrangement of cell adhesion via E-Cadherin re-distribution controls the branching process. Periodic activation of Notch signaling drives the differential cell adhesion and contraction of basal filopodia, which occurs only below an FGF signaling threshold. Our results suggest a coordinated adjustment of cell shape and adhesion orchestrates feather branching, which is regulated by Notch and FGF signaling.

---

[1] Institute of Life Sciences, Fuzhou University, Fuzhou, Fujian 350116, China. [2] Department of Mathematics, Fuzhou University, Fuzhou, Fujian 350116, China. [3] National Institute of Biological Sciences (NIBS), Beijing 102206, China. [4] Department of Pathology, University of Southern California, Los Angeles, CA 90033, USA. These authors contributed equally: Dongyang Cheng, Xiaoli Yan, Guofu Qiu, Juan Zhang. Correspondence and requests for materials should be addressed to Z.Y. (email: zyue@fzu.edu.cn)

Biological systems utilize various principles to achieve periodic pattern formation[1–4]. Periodic epithelial branching is a widely used mechanism to increase the surface area of an organ. Such a mechanism is exemplified in feather branching, which characterizes modern birds[5,6]. In this process, the epithelial sheath at the base of the follicle organizes into periodic branches[7–11] (Fig. 1). Recently, the regularly branched feather structure was utilized as a model to dissect the pathological principles of tissue damage due to chemo- and radiation therapy, because any perturbations of feather development are recorded in the final feather morphology[12–15]. Thus, the formation of the exquisite feather branches is of both evolutionary and medical interest.

Feather branching has been considered as a classical example of how periodic structures result from the reaction-diffusion mechanism during pattern formation[10]. The involvement of the antagonistic molecule pairs such as BMP4/Noggin and BMP2/Shh has been proposed[10,11]. Furthermore, a set of core signaling molecules, including BMP, Shh, Wnt and FGF, has been shown to regulate this process[9–11,16–18]. However, it remains unclear at the cell level how the keratinocytes are organized into the periodic branches.

Here we report how cells accommodate the rapid formation of feather branches through the rearrangement of cell adhesion and changes in cell shape, and how molecular signaling controls the patterning of the periodic feather branches. We find that extensive filopodia present on basal keratinocytes before branching, which disappear after branch formation. These filopodia are regulated by the Rho family small GTPases *RhoA* and *Cdc42*, and help interpret the FGF signaling gradient in the feather follicle. FGF and Notch signaling regulate the branching process and further control the formation of the filopodia. Calculating the surface area before and after branching reveals a scaling effect resembling the "coastline paradox", which was proposed by Benoit Mandelbrot in the 1960s to describe the fractal nature of the coastline[19]. Thus counter-intuitively, the surface area increase during feather branching morphogenesis is actually prepared in advance. These results provide mechanistic insight into the epithelial branching process.

## Results

### Filopodia in basal keratinocytes of the feather epithelium.
We examined the ultrastructure of feather epithelium before and after branching (Fig. 2a, b; and Supplementary Fig. 1). Transmission electron microscopy (TEM) analysis revealed extensive filopodia in basal keratinocytes in the pre-branch feather epithelium (Fig. 2c). Higher magnification views showed clear basal lamina along the filopodia, including the lamina densa and lamina lucida (Fig. 2c). Depending on the specific location in the feather follicle, these filopodia vary in size and length. On average, each basal cell extends 3–5 filopodia about 2–10 μm long as counted/measured from the TEM images, with no single filopodium showing dominance over the others. Filopodia from two neighboring cells may fuse together, with the cell membranes running side by side to separate the cells (Supplementary Fig. 1a). Upon branching, the filopodia disappear and a smooth basal lamina is formed. Still, adjacent basal keratinocytes form tight junctions (TJs) in the apical/basolateral border, and zone of adherens junctions (AJs) at the sites of lateral cell-cell contact (Fig. 2d and insert). Therefore, even with the extensive filopodia, the basal keratinocytes retain these classical adhesion structures.

We characterize the filopodia by additional marker analysis (Fig. 2e). FITC-phalloidin showed strong staining in the filopodia, suggesting the presence of rich F-Actin bundles. E-Cadherin and β-Catenin are also expressed. When double-stained with a mesenchymal marker Tenascin C (Tn), we found inter-

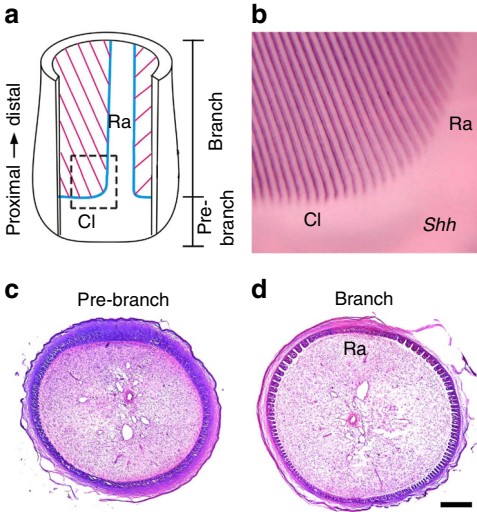

**Fig. 1** Feather branching morphogenesis. **a** A schematic diagram showing the developing feather follicle. The proximal follicle epithelium called the collar (Cl) is not branched. Above the collar, the feather branches along the circumference of the follicle, except in the rachis (Ra). The dashed box is enlarged in **b**. **b** Feather branching is marked by *Shh* in situ hybridization. **c**, **d** Cross sections of the developing feather follicle in the lower collar level (pre-branch) and the upper branched level. The rachis (Ra) region is not branched. Bar = 100 μm

digitation of β-Catenin and Tn staining, suggesting the filopodia project into the mesenchyme and do not result from artificial contraction of the epithelium or mesenchyme during sample preparation. Finally, VASP and Fscn1 are markers normally associated with filopodia; they also stained positive in these structures (Fig. 2e and Supplementary Fig. 2).

### Contraction of filopodia controls feather branching.
Filopodia are cell projections that are subject to the regulation of Rho family small GTPases. We tested whether these molecules can regulate the filopodia in basal keratinocytes. We cloned both the constitutively active (CA) and dominant negative (DN) forms of *RhoA*, *Rac1*, and *Cdc42* into lentivirus. The capability of these constructs to regulate filopodia was confirmed in vitro (Supplementary Fig. 3). In vivo, two independent methods were used to examine the roles of these genes in feather development (Fig. 3a, b; Supplementary Fig. 4): In the first method, we directly injected lentivirus into actively growing feather follicles and examined the impact of local gene perturbation[14,15]; In the second method, we made transgenic feathers via lentiviral-mediated overexpression or RNAi knockdown[18]. The CA forms did not disrupt feather formation, while the DN forms of *RhoA* and *Cdc42* induced ectopic branches in the rachis, and loss of rachis in the feathers (Fig. 3c, d). DN-*Rac1* produced normal feathers, consistent with its inability to disrupt filopodia in cell culture.

Because the DN forms of GTPases may elicit non-specific effects[20,21], and there are over 20 Rho family GTPases in the avian genome (Supplementary Table 1)[22,23], we further verified the impact of RNAi knockdown of these small GTPases. The knockdown efficiency and specificity of RNAi were each verified in vitro and in vivo (Supplementary Figs. 4 and 5). In particular, RNAi for *RhoA* or *Cdc42* did not perturb the expression of other Rho family GTPases (Supplementary Fig. 5). Consistent with the results from overexpressing the DN forms, knockdown of *RhoA* or *Cdc42* produced feathers with weaker or no rachis in the upper part, whereas knockdown of *Rac1* resulted in normal feathers

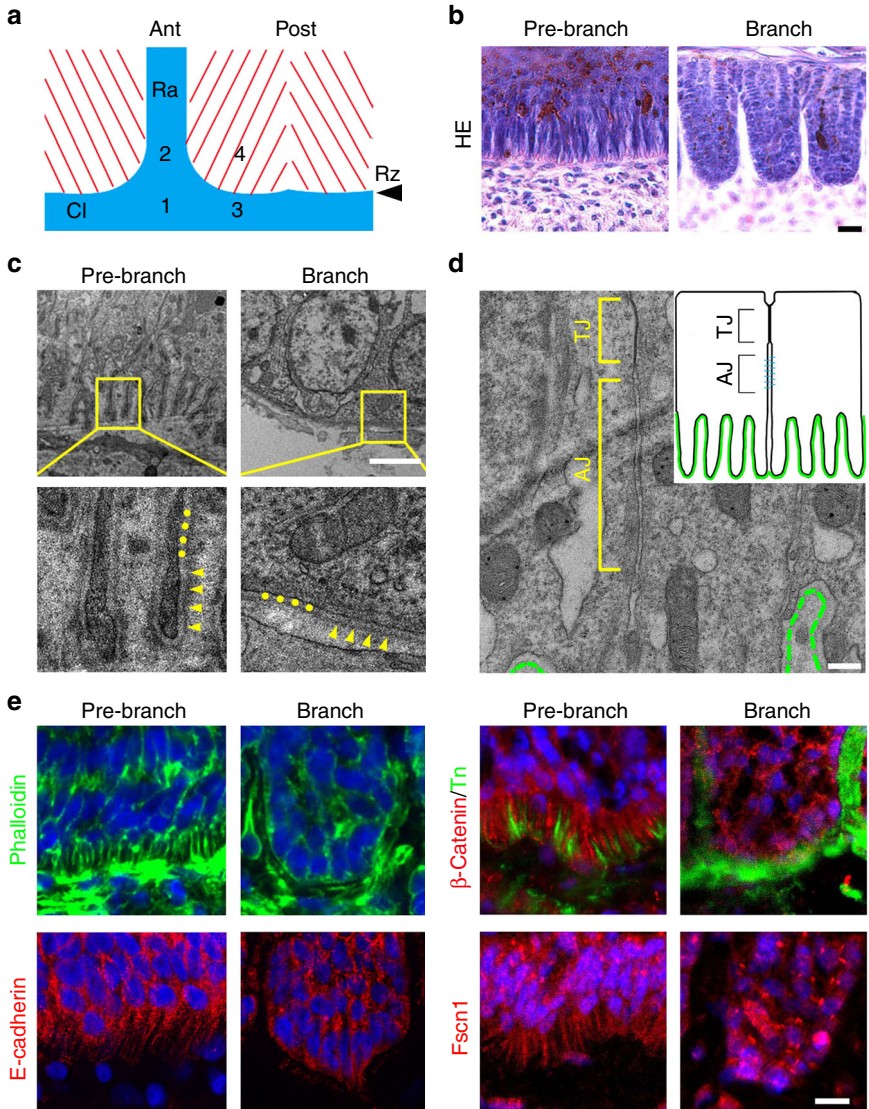

**Fig. 2** Filopodia in basal keratinocytes of the feather epithelium. **a** A schematic diagram showing the regions examined. The proximal feather epithelium (collar, Cl; regions 1, 3) is not branched. Above the ramogenic zone (Rz), the epithelium branches along the circumference of the follicle, with the exception of the rachis in the anterior (Ra, region 2). Filopodia were found in regions 1, 2, 3 but not 4. **b** HE staining; **c**, **d** TEM analysis; **e** immunofluorescence showing filopodia in the basal keratinocytes of the pre-branch feather epithelium. Basal lamina are shown in higher magnifications in **c**, with the lamina lucida indicated by dots, and lamina densa indicated by arrow heads. Tight junction (TJ) and adherens junctions (AJ) are indicated in the basal keratinocytes (**d** and inset). Dashed green line indicates the basal lamina. Representative images from five repeated experiments are shown. Bar = 20 μm in **b**, **e**, 2 μm in **c**, 0.3 μm in **d**

(Fig. 3e). These results demonstrate that the regulation of basal filopodia is causally linked with feather branching morphogenesis.

**E-Cadherin regulates feather branching**. A few possibilities may explain, at the cell level, how exactly the feather epithelium is organized into branches: differential proliferation, differential cell death/apoptosis, and differential cell adhesion. No pre-patterned cell proliferation or apoptosis was found before or immediately after branch formation, as shown by TEM analysis, PCNA staining, or TUNEL staining (Fig. 4a, b and Supplementary Fig. 6). We reasoned that cells are directly re-arranged into the periodic pattern, possibly through a differential adhesion mechanism. Consistently, we found that there are higher levels of

E-Cadherin and β-Catenin in each barb plate[24], while the marginal plate cells express lower levels of these molecules (Fig. 4c, d).

Because E-Cadherin-mediated cell adhesion depends on its organization at the nanoscale[25], we examined in detail its distribution pre- and post- feather branching. In the pre-branch basal keratinocytes, AJs were formed between two adjacent cells; however, it appears the E-Cadherin molecules were more diffusively distributed (Fig. 4e, f). On the other hand, in the branched feather barbs, the outer layer cells (marginal plate) use more stable TJs and desmosomes to build cell connections, as these cells have reduced levels of E-Cadherin (Supplementary Fig. 1b). The inner barb cells, which have higher levels of E-Cadherin, showed distinct puncta of E-Cadherin distribution (Fig. 4e), and a continuous zone of AJs under TEM (Fig. 4f). These structures resemble the previously described adhesion

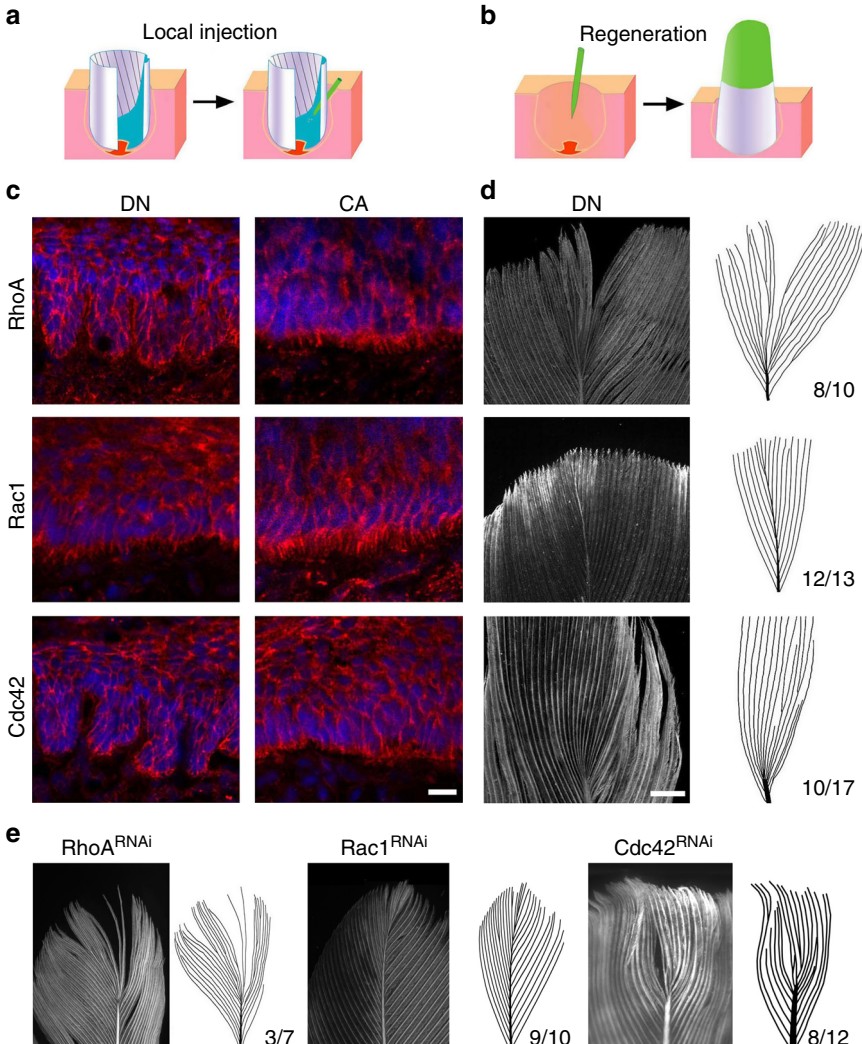

**Fig. 3** Filopodia regulate feather branching. **a**, **b** Schematics showing two different methods for manipulating gene expression in the feather follicle. Lentivirus were either directly injected into the developing follicle (**a**), or into the follicle cavity after plucking the feather (**b**). **c**, **d** The dominant negative (DN) forms of *RhoA* and *Cdc42*, but not *Rac1* nor their constitutively active (CA) forms regulated filopodia in vivo as shown by β-Catenin staining (**c**, local injection in the rachis region) and the final feather form (**d**, transgenic feathers). **e** RNAi knockdown of *RhoA* or *Cdc42*, but not *Rac1* disrupted feather axis formation. The numbers indicate the occasions of feathers with the phenotype. Bar = 20 µm in **c**, 2 mm in **d**, **e**

zipper structure[26]. Thus, E-Cadherin is down-regulated in the basal keratinocytes and re-distributed in the suprabasal cells during feather branching.

We tested the functional significance of E-Cadherin-mediated cell adhesion in feather branching morphogenesis. When *E-Cadherin* was ectopically overexpressed, the branching of feather epithelium was blocked; conversely, we obtained feathers with supernumerary branches and reduced rachis size when *E-Cadherin* expression was suppressed (Fig. 4g, h). Therefore, differential epithelial cell adhesion via redistribution of E-Cadherin is required for feather branching.

**Activation of Notch signaling drives feather branching**. We then explored the molecular pathways that control the feather branching process. Notch signaling is often harnessed to generate periodic spatial patterns and has been implicated in embryonic feather development[27–29]. In our effort to profile gene expression in the adult feather follicle[18], we identified members of the Notch signaling pathway including *Notch1*, *Notch2, Serrate1, Serrate2* (Supplementary Table 1). Here we mapped the expression of

these genes during feather branching (Fig. 5a–d and Supplementary Fig. 7): *Notch1* and *Serrate1* are enriched in the pre-branch feather epithelium but are expressed at low levels in the basal keratinocytes. *Notch2* expression is more ubiquitous, whereas *Serrate2* is mainly expressed in the basal keratinocytes. After branching, *Notch1* is enriched in the barb plate, *Serrate2* is enriched in the complementary marginal plate, whereas *Notch2* and *Serrate1* are more ubiquitously expressed.

The role of Notch signaling in feather development is examined in vivo. RNAi-*Notch1* (and *Notch2*) produced feathers with supernumerary branches (Fig. 5e–h). Similar results were obtained for RNAi-*Serrate1* and RNAi-*Serrate2* (Supplementary Fig. 8). In contrast, overexpression of *Notch1* resulted in barb fusion and the formation of multiple rachises. The specificity and knockdown efficiency for each RNAi construct were examined in vitro and in vivo (Supplementary Figs. 4 and 5). Moreover, mis-expression of *Delta1* in the feather follicle also disrupted the regular branched pattern (Supplementary Fig. 8), further supporting the involvement of Notch signaling in feather branching.

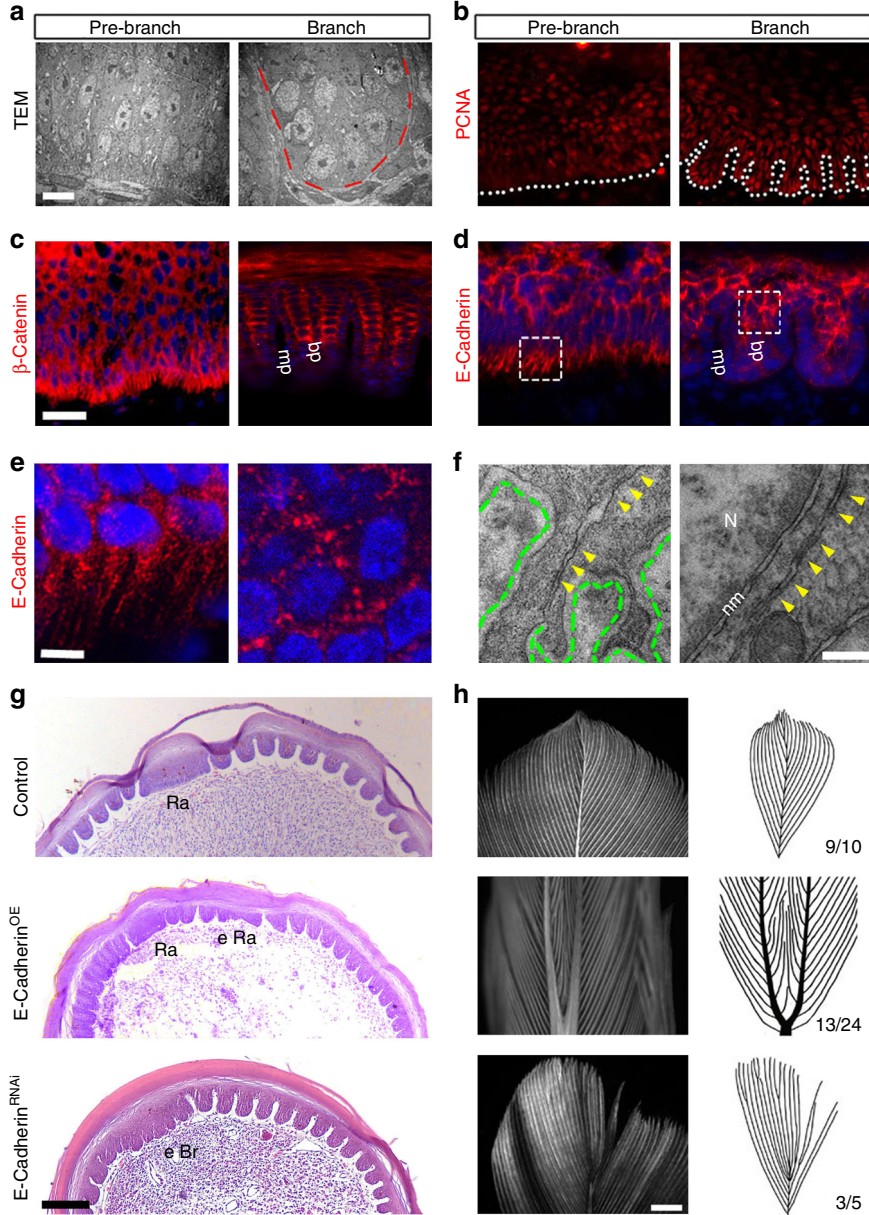

**Fig. 4** Topological arrangement of E-Cadherin-mediated cell adhesion is required for feather branching. **a, b** Neither pre-patterned cell proliferation nor apoptosis play a role in feather branching morphogenesis as shown by TEM analysis and PCNA staining. **c, d** Differential distribution of β-Catenin and E-Cadherin in feather branching. mp, marginal plate; bp, barb plate. **e, f** Higher magnification views of regions in **d** showing E-Cadherin was diffusely distributed in pre-branch feather epithelium, but as puncta in branched barbs (**e**). Arrow heads indicate unstable adherens junctions (AJ) in the filopodia, as compare to AJ clusters in branched barbs (**f**). Dashed green lines indicate the basal lamina. Representative images from five repeated experiments are shown. N, nucleus; nm, nuclear membrane. **g, h** Overexpression or knockdown of *E-Cadherin* disrupted feather branching. Histology and gross feather morphology are shown. The numbers indicate the occasions of feathers with the phenotype. Ra, rachis; eRa, ectopic rachis; eBr, ectopic branch. Bar = 2 μm in **a**, 20 μm in **b-d** (shown in **c**), 5 μm in **e**, 0.3 μm in **f**, 100 μm in **g**, 2 mm in **h**

Given the complexity of Notch signaling activation[4,30–32], we examined the activity of this pathway in the feather follicle. Three independent criteria were explored. First, we examined the expression patterns of down-stream Notch target genes including *L-Fringe* and *Hey1*[33,34]. In situ hybridization showed that they both are expressed in the marginal plate (Fig. 5i, j; Supplementary Fig. 7). Second, we cloned a Notch reporter into lentivirus, where GFP expression was driven by a promoter containing 6 × RBP-J binding elements[35,36]. In the developing feather follicle, GFP is only expressed in the marginal plate, indicating specific activation of Notch signaling in this region. For the control, a viral vector where GFP expression was driven by a CMV promoter showed widespread expression (Fig. 5k, l). Finally, we constructed a secretory form of Serrate2 (sSer2; Supplementary Fig. 9a), which is known to inhibit Notch signaling[37,38]. We demonstrated sSer2 can reduce the Notch reporter activity in cell culture (Supplementary Fig. 9b). When overexpressed in vivo, sSer2 induced ectopic rachis formation (Fig. 5m). Altogether, these results suggest that activation of Notch signaling is required for the periodic feather branching.

High Notch1 expression levels in the barb plate may orchestrate the redistribution of E-Cadherin and β-Catenin, as

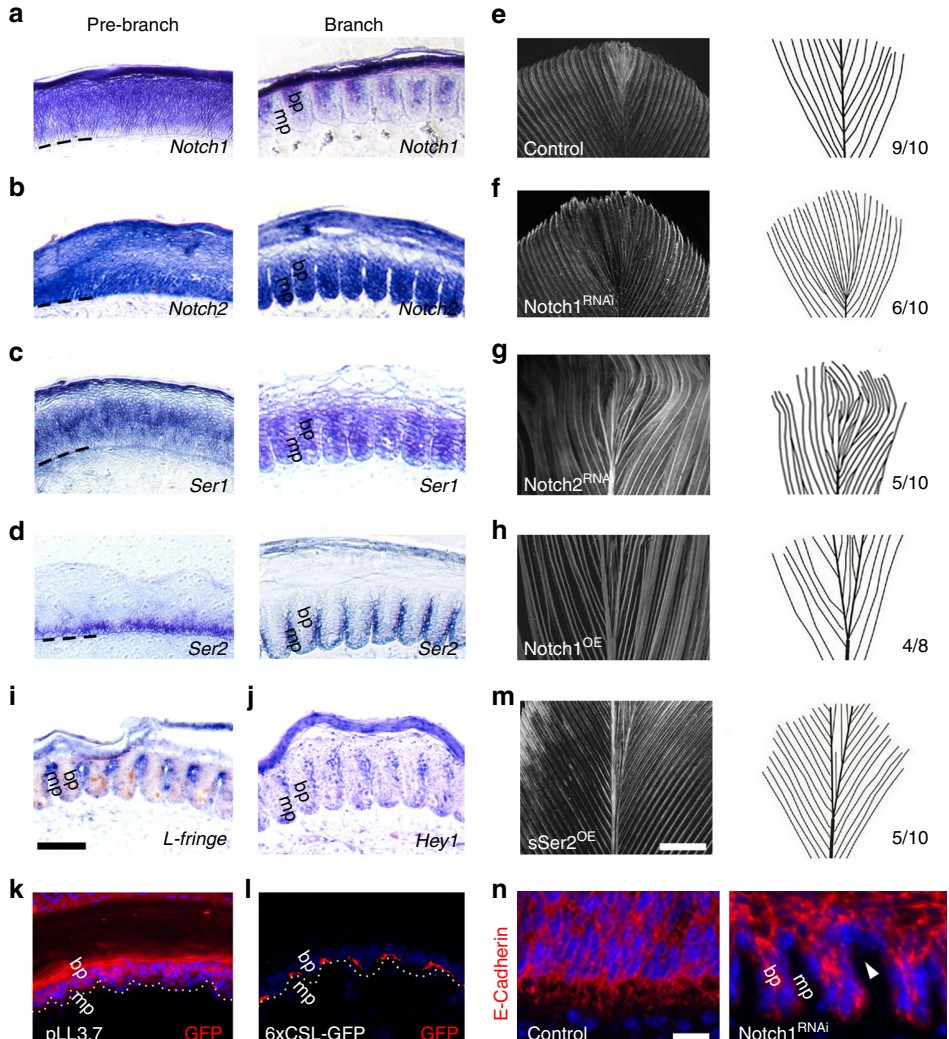

**Fig. 5** Notch signaling regulates feather branching and the contraction of filopodia. **a–d** The expression patterns of *Notch1*, *Notch2*, *Ser1*, *Ser2* in feather branching. **e–h** Phenotypes of *Notch1* and *Notch2* manipulation in the feather follicle. The numbers indicate the occasions of feathers with the phenotype. **i, j** The expression patterns of downstream target genes *L-fringe* and *Hey1* in the feather follicle. **k, l** Expression of a 6XCSL-GFP reporter was specifically in the marginal plate of branching feather epithelium, whereas a control viral vector showed widespread virus expression in the follicle. **m** A secretory form of *Ser2* (*sSer2*) induced ectopic rachis formation. **n** Ectopic activation of Notch signaling via local injection of RNAi-*Notch1* lentivirus led to filopodia contraction and E-Cadherin down-regulation in the marginal plate cells (arrow head). Representative images from three repeated experiments are shown. mp, marginal plate; bp, barb plate. Bar = 100 μm in **a–d** and **i–l** (shown in **i**); 2 mm in **e–h** and **m** (shown in **m**); 20 μm in **n**

it directly binds β-Catenin in Drosophila[29] and in vertebrate cells (Supplementary Fig. 10a). In addition, activation of Notch signaling down-regulates E-Cadherin expression (Supplementary Fig. 10b), a regulatory module often found in cancer metastasis. Furthermore, activation of Notch signaling reduces cell filopodia (Supplementary Fig. 10c). Indeed, when Notch signaling was ectopically activated in the rachis in vivo, we found reduced E-Cadherin and contraction of filopodia in the ectopic branches (Fig. 5n). In summary, the periodic activation of Notch signaling may drive the differential cell adhesion and contraction of basal filopodia, leading to feather branching.

**Filopodia help interpret the proximal-distal FGF gradient**. The fact that the feather epithelium branches only at a distance from the most proximal anchoring site, the dermal papilla (dp), suggests a morphogen gradient is in control. We have shown previously a proximal-distal FGF signaling gradient regulates feather branching[17]. Indeed, FGF2 and FGF10 showed a graded

distribution pattern in the feather follicle (Fig. 6a–c). Perturbation of FGF signaling by local injection of RNAi-*FGFR1* or through a specific chemical inhibitor SU5402 induced ectopic branches in the rachis region (Fig. 6d). Furthermore, FGF signaling also regulates the filopodia in basal keratinocytes. SU5402 treatment resulted in ectopic branching and the filopodia disappeared (Fig. 7a). In contrast, implantation of FGF10-soaked beads blocked epithelial branching and induced ectopic rachis formation: in the ectopic rachis, filopodia were also induced (Fig. 7b). These data are consistent with a role of FGF signaling in the regulation of cell filopodia in vitro (Supplementary Fig. 11). Conversely, we found that the filopodia can sense FGF molecules as they stained positive for FGFR1 and can transport FGF10 molecules (Fig. 7c). TEM analysis further documented vesicle-like structures inside the filopodia, supporting its role in transportation (Fig. 7d). Thus, a positive feedback loop may exist between FGF signaling and the filopodia, which helps to interpret the proximal-distal FGF gradient in the feather follicle.

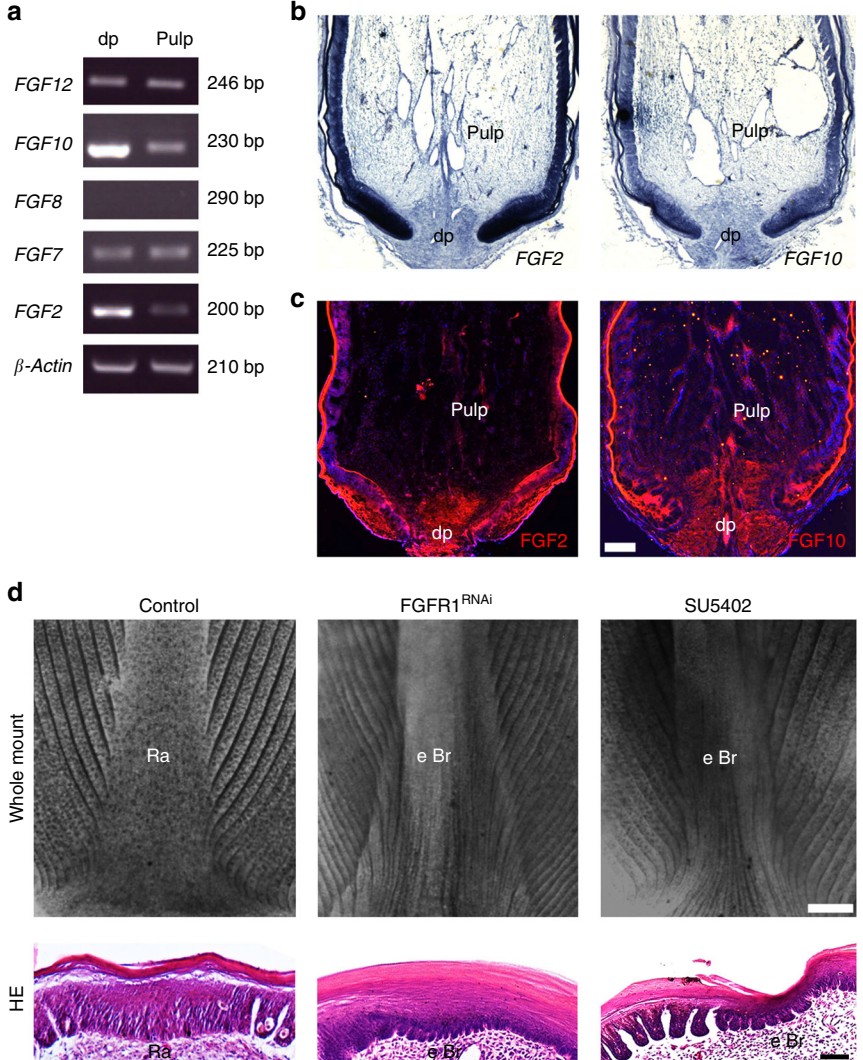

**Fig. 6** A proximal-distal FGF signaling gradient regulates feather branching. **a** Semi-quantitative RT-PCR analysis revealed graded expression of *FGF10* and *FGF2* in the feather follicle. The dermal papillae (dp) and the pulp represent the proximal and distal compartment of the follicle, respectively. **b** In situ hybridization and **c** immunofluorescence showing the graded expression of FGF2 and FGF10 in the feather follicle. **d** Whole mount view and HE sections of the feather rachis after manipulating the FGF signaling via RNAi-*FGFR1* lentivirus local injection, or SU5402 (100 μM) chemical inhibitor treatment in vivo. Ectopic branches were induced by both treatments. Representative images from five repeated experiments are shown. Ra rachis, eBr ectopic branch. Bar = 100 μm

**Filopodia and the surface area change in feather branching.** Since feather branching is coupled with the disappearance of basal filopodia, we wonder how the total surface area actually changes during this process. We designed an algorithm to delineate the epithelial-mesenchymal (E-M) border in the image, and calculated the ratio of surface area increase due to the filopodia or branch formation (simplified to 2D situation, the length of the E-M border $\lambda$ versus the linear distance $d$)[39]. The $\lambda/d$ ratio is in the range of 5–16 and averaged 11 before branching. After branching, $\lambda/d$ is in the range of 4–8 and averaged 6 (Fig. 7e). Therefore, the total surface area is actually decreased by about 50% after feather branching. This situation resembles the coastline paradox[19], which claims that a given landmass may not have a fixed coastline length because of the fractal-like property of its coastline. Thus at the nanoscale, emerging features - in this case basal filopodia, increase the surface area of the pre-branch feather epithelium.

**Discussion**

We propose a model that integrates the molecular and cellular events in feather branching morphogenesis (Fig. 7f): A proximal-distal gradient of FGF signaling cooperates with periodic Notch activation to regulate feather branching. Notch signaling is activated only when FGF levels fall below a threshold. The periodically activated Notch signaling then drives the contraction of filopodia and differential cell adhesion, promoting branch formation.

The basement membrane of an epithelial tissue is often viewed as a flat sheet where basal keratinocytes attach. This is not true even for mammalian skin: in the mouse footpad, basal keratinocytes also have filopodia about 0.5–1 μm long, and the basal lamina follow the outline of the filopodia (Supplementary Fig. 1c). These structures are particularly distinct in the feather follicle, possibly due to the intensive epithelial-mesenchymal interactions in feather branching morphogenesis. Similar elongated cyto-projections have been implicated in several examples of organ

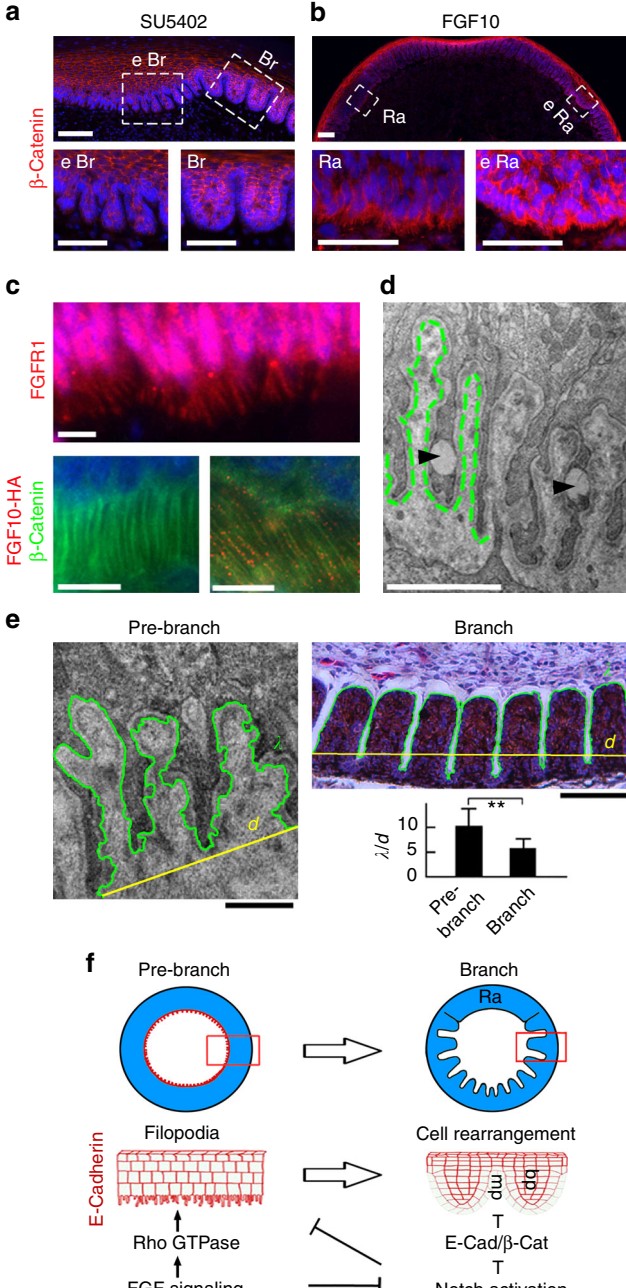

**Fig. 7** The filopodia sense and transport FGF10 and increase the surface area for feather branching. **a** SU5402 induced ectopic branch (eBr) in the rachis and inhibited filopodia. **b** Locally implanted FGF10 beads inhibited branching and induced filopodia in basal keratinocytes. **c** Filopodia expressed FGFR1, and transported FGF10. Notice the FGF10 puncta (red) were lined up along the filopodia (green). **d** A TEM image showing the transported vesicles inside the filopodia (marked by arrowheads). Dashed green line indicated the basal lamina. **e** The circumference (λ) and distance (d) of the epithelial—mesenchymal border were measured, and their ratio was calculated. Ten follicles were analyzed and representative images are shown. **, $p < 0.01$ by $t$-test. **f** Schematics showing FGF signaling cooperates with periodic Notch activation to regulate the filopodia and E-Cadherin/β-Catenin-mediated cell adhesion, which then control the periodic feather branching. Bar = 50 μm in **a**, **b**, **e**; 10 μm in **c**; 2 μm in the TEM images

development, such as the cytoneme[40,41]. A wide range of morphogens have been shown to be transported by these long projections including Delta, Hedgehog, Dpp, and Wnt[40–43]. In the feather follicle, the filopodia sense and transport FGF10 molecules, which may help interpret the proximal-distal FGF gradient.

The feather branching process is accompanied by a complete contraction of the basal filopodia. We have shown that by regulating the filopodia via small GTPases, the feather branching pattern was perturbed. Thus the contraction of filopodia is causally linked with branching morphogenesis. The positive feedback loop between FGF signaling and filopodia may contribute to this abrupt contraction. Additionally, Notch signaling also contributes to the contraction of filopodia, as demonstrated by our in vivo manipulation of this pathway. Filopodia contraction may facilitate Notch pathway activation and feather branching by reducing FGF signaling.

Given the complex expression patterns of the various Notch ligands and receptors in the feather follicle, and the potential *cis* and *trans*-interactions of the ligands/receptors[4,30–32], the activation of Notch signaling in the feather follicle is likely to be complicated. We have examined the impact of overexpression and knock-down of both the receptors (*Notch1*, *Notch2*) and ligands (*Ser1*, *Ser2*). It appears Notch2 is the endogenous receptor that is responsible for the activation of Notch signaling in the marginal plate keratinocytes. Ser1 serves as the ligand to drive its activation, whereas Ser2 acts in *cis* to inhibit its activation. The expression of L-Fringe may further modulate Notch activation[44]. Our results are consistent with the current understanding of Notch signaling activation[4,30–32]: overexpression of Notch receptors inhibits branching, because limited amounts of endogenous ligands (Ser1, Ser2) will be sequestered in *cis*, thus reducing Notch signaling *trans*-activation. Conversely, suppression of Notch receptors will render more available ligands for Notch signaling *trans*-activation. Similarly, because Ser1/2 is inhibitory in *cis*, down-regulation of these molecules will promote Notch activation and branch formation.

The periodic branching of feather epithelium is an example of the classical reaction-diffusion mechanism in pattern formation[10,11]. Here we propose that a proximal-distal FGF gradient cooperates with periodic activation of Notch signaling to control this process. Both in vivo (Fig. 7b) and in vitro (Supplementary Fig. 10d), FGF signaling inhibits Notch activation. At the cell level, contraction of filopodia and rearrangement of E-Cadherin-mediated cell adhesion is critical for feather branching. Since filopodia can actively sense and transport FGF molecules, they may alter the fate of the basal keratinocytes so they are competent to branch. Toward the base of the follicle, FGF levels are high and inhibit branching. In contrast, activation of Notch signaling in the distal feather drives branching and reduces the filopodia, and the keratinocytes are more tightly compacted/connected. In this sense, a pair of "activator–inhibitor" molecules still work together to control the cell status and feather branching, although not in the classic manner of "reaction–diffusion".

Branching morphogenesis is widely used in many organs to increase the surface area. Counter-intuitively, we show here that the tissue actually prepares the surface area in advance, via filopodia, to accommodate feather branching. The total surface area is reduced immediately after feather branching, although cell proliferation may further increase the surface area in later feather growth. Our data illustrate how complex molecular activities and cell behaviors are integrated to control periodic pattern formation in feather development.

## Methods

**Feather follicle manipulation in vivo**. Three to six months adult male chickens (Gallus gallus domesticus) were purchased from a local farm and housed in Fuzhou

University Animal Facility Center. All experiments were approved by the Animal Research Committee in Fuzhou University. The chickens were anesthetized using pentobarbital (intraperitoneal injection, 50 mg kg$^{-1}$) before surgery. For lentiviral-mediated gene overexpression and RNAi knockdown, fully grown primary wing feathers were plucked and 200 µl virus solutions were injected into the follicle cavity using a micropipette. Feathers were allowed to regenerate for one month before sample collection to document the gross morphology using a stereo dissection microscope (Chongqing Optical Instrument, China). For local injection of virus/protein/chemical reagents, contour feathers in the wing in their actively growing phase were used. Two to five microliter solutions were injected into the desired locations in the follicle using a homemade glass microneedle. FGF10 (0.1 µg µl$^{-1}$, Sangon, Shanghai, China) or SU5402 (100 µM, Santa Cruz, Dallas, Texas, USA) was mixed with Sepharose 4B beads in PBS before injection. Samples were collected 48 h later and fixed in 4% paraformaldehyde (PFA) before processing for documentation of the gross morphology or histology[13].

**Cell culture and transfection**. All cells were purchased from the Cell Library of the Chinese Academy of Sciences, Shanghai, China (293 T cells, #GNHu17; DF-1 cells, #GNO30; HeLa cells, #TCHu187; MCF7 cells, #TCHu74). Cells were cultured with 10% fetal calf serum (Hyclone, Xiamen, China) in DMEM (Life Technology, Guangzhou, China). Cells were maintained in a humid incubator at 37 °C with 5% CO$_2$. Plasmids were transfected using calcium phosphate (for 293 T cells) or electroporation (for DF-1/HeLa/MCF7 cells). A home-made electroporator was used for electroporation (680 V/30 ms for 1 pulse, 10 µg plasmids mixed with $4 \times 10^6$ cells in 120 µl total volume). Cells were lysed for total RNA extraction (DF-1 cells) or sonicated for protein collection (293T cells) using the standard protocols.

**Lentiviral construction**. We used the pLL3.7 vector for short hairpin RNAi knockdown. Target sequences for RNAi were listed in Supplementary Table 2, and a scramble control was used[18]. The vector used for gene overexpression was pLVX-ZsGreen. Source of the genes: full-length human E-CADHERIN was purchased from SinoBiologicals, Beijing, China; human SERRATE2 on pCIG (a generous gift from Dr Fernando Giraldez, Universitat Pompeu Fabra, Spain) was digested with BamHI to achieve the secretory form (amino acid 1-994); full length mouse Notch1 was a generous gift from Dr Olivier Pourquie, IGBMC, France. Lentivirus was packaged in 293T cells using standard protocols.

**RNAi knockdown efficiency**. To examine the RNAi knockdown efficiency, the lentiviral constructs were electroporated into DF-1 cells (which is a chicken fibroblast cell line) and total RNAs were extracted 48 h later. The full-length chicken E-Cadherin cDNA was cloned into the pEGFP-N1 expression plasmid and co-electroporated with the RNAi construct. To examine the knockdown efficiency in vivo, virus infection was performed in plucked feather follicles and samples were collected 4 days post-infection. Each follicle was individually collected and total RNAs extracted for qRT-PCR analysis. Primer sequences were listed in Supplemental Table 2.

**FGF10-HA fusion protein**. Full length FGF10 was PCR cloned from a chicken cDNA library and fused with an HA-tag in the C-terminus on a pcDNA6.0 vector. FGF10-HA was expressed in 293 T cells, lysed by sonication, centrifuged at 9600 × g for 1 min to collect the supernatant and aliquoted. The supernatant was injected into the feather follicle, and samples were collected 24 h later.

**Immunofluorescence and in situ hybridization**. Standard histological methods were followed. Fluorescence images were acquired using a Leica confocal microscope. Antibodies and reagents used: FITC-Phalloidin (Sigma, Shanghai, China; P5282, 5 µg ml$^{-1}$ in PBS), β-Catenin (Sigma, Shanghai, China; C2206, 1:200 dilution), PCNA (Santa Cruz, Dallas, USA; sc-7907, 1:200 dilution), Fscn1 (Sangon, Shanghai, China; D120251, 1:200 dilution), VASP (Sangon, Shanghai, China; D124097, 1:200 dilution), FGF2 (Sangon, Shanghai, China; D160122, 1:200 dilution), FGF10 (Sangon, Shanghai, China; D163308, 1:200 dilution), FGFR1 (Sangon, Shanghai, China; D120628, 1:200 dilution), GFP (Beyotimes, Shanghai, China; AF0159, 1:200 dilution), HA-tag (Beyotimes, Shanghai, China; AF0039, 1:200 dilution), E-Cadherin (Developmental Studies Hybridoma Bank; 7D6, 1:40 dilution), Tenascin C (Developmental Studies Hybridoma Bank; M1-B4, 1:40 dilution), BrdU (Developmental Studies Hybridoma Bank; G3G4, 1: 40 dilution). For counter staining with 4′,6-diamidino-2-phenylindole (DAPI), the sections were mounted with DAPI (0.5 µg ml$^{-1}$) in the mounting medium (30% glycerol in PBS). For TUNEL staining, we used a commercial kit from Beyotimes (Shanghai, China) and the manufacturer's instructions were followed. cRNA probes used for in situ hybridization were listed in Supplementary Table 2.

**RNA extraction and RNA-seq**. Total RNAs were extracted from the whole feather follicle or the dissected dermal papilla and mesenchymal pulp. The standard Trizol protocol was followed. Total RNAs were reverse transcribed using the HiScript Q RT Supermix (Vanzyme, Nanjing, China). Semi-quantitative PCR was performed on a 2720 Thermal Cycler, Applied Biosystems. Quantitative PCR was performed

on Roche LightCycler 480. Primers sequences were listed in Supplementary Table 2. RNA-seq was performed on the Illunina HiSeq 4000 platform (Novogene, Beijing, China), and data were mapped to the Galgal4 genome assembly.

**Transmission electron microscopy**. Freshly dissected tissue samples from different regions in the feather follicle were fixed in primary fixative (0.2% Tannic Acid, 3% Glutaraldehyde in MOPS buffer, pH7.0) and secondary fixative (1% OsO4 in water), stained with 2% Ur-Acetate, and mounted in resin. Ultrathin sections were collected and examined using an FIE Tecnai G2 Spirit Transmission Electron Microscopy (TEM).

**Western blot and immunoprecipitation**. For Western blot analysis, MCF7 cells were electroporated with the vector (pEGFP-N1), NICD or the full-length Notch1 expression plasmids. Cells were lysed 48 h later to detect E-Cadherin and β-Catenin levels. β-Actin was used as a control for equal loading. The relative expression levels were quantified using the ImageJ program. For immunoprecipitation, 293T cells were transfected with a Notch1 expression plasmid (which has a Myc-tag in the C-terminus), and cells were lysed and immunoprecipitated by using an anti-Myc antibody (Sangon, Shanghai, China; D110006, 1:50 dilution; for WB, 1:1000 dilution). Normal rabbit IgG was used as a control. Other antibodies: Cdh1 (Sangon, Shanghai, China; D160656, 1:1000 dilution), β-Catenin (Sigma, Shanghai, China; C2206, 1:1000 dilution), Actin (Sangon, Shanghai, China; D110001, 1:1000 dilution), Full-length gel images can be found in Supplementary Fig. 12. Representative results from three repeats were shown.

**Luciferase reporter assay**. In total 293 T cells were subcultured in 24-well plate 24 h before transfection. Cells were transfected with a 6XCSL Notch reporter plasmid, with or without sSer2 or FGF10 co-transfection. Luciferase activity were measured 48 h post-transfection. For control, the small molecular inhibitor of Notch signaling N-[N-(3,5-Difluorophenacetyl)-L-alanyl]- S-phenylglycine t-butyl ester (DAPT) was added to the culture medium at a final concentration of 46 nM 24 h before sample collection. Each experiment was repeated at least three times and representative results were shown.

**Image processing**. We used a region-based active contour model (ACM) to perform image segmentation and locate the boundaries of objects[39]. This method transforms image segmentation to an energy minimization problem, where the energy function is defined on a dynamic curve which achieves its minimum when the curve conforms to the boundary. Given an image $I(x):\Omega \rightarrow \Re$, where $\Omega \subset \Re^2$ is the image domain. The model implements segmentation by finding an evolution curve $C \subset \Omega$ that minimizes the following energy function:

$$
\begin{aligned}
F(\phi, c_1, c_2, f_1, f_2) = {} & \lambda_1 \int_\Omega |I - c_1|^2 H(\phi(x))\mathrm{d}x \\
& + \lambda_2 \int_\Omega |I - c_2|^2 (1 - H(\phi(x)))\mathrm{d}x \\
& + \eta_1 \int \left( \int_\Omega K_\sigma(x-y)|I(y) - f_1(x)|^2 H(\phi(y))\mathrm{d}y \right)\mathrm{d}x \\
& + \eta_2 \int \left( \int_\Omega K_\sigma(x-y)|I(y) - f_2(x)|^2 (1 - H(\phi(y)))\mathrm{d}y \right)\mathrm{d}x \\
& + \nu \int \delta(\phi)|\nabla \phi(x)|\mathrm{d}x
\end{aligned}
\tag{1}
$$

where $H(\cdot)$ denotes the Heaviside function, and $\delta(x) = \frac{\mathrm{d}}{\mathrm{d}x} H(x)$ is the Dirac function. $K_\sigma$ is a Gaussian kernel with the standard deviation $\sigma > 0$, which is defined as $K_\sigma(x) = \frac{1}{2\pi\sigma^2} e^{-|x|^2/2\sigma^2}$. The Heaviside function and Dirac function is approximated as

$$
H_\varepsilon(x) = \frac{1}{2}\left(1 + \frac{2}{\pi}\arctan\left(\frac{x}{\varepsilon}\right)\right) \text{ and } \delta_\varepsilon(x) = \frac{1}{\pi}\frac{\varepsilon}{\varepsilon^2 + x^2}, 0 < \varepsilon \ll 1
$$

Applying the gradient descent method to Eq. (1), the optimal values of $c_1, c_2, f_1, f_2$ for minimizing the energy functional defined by Eq. (1) can be achieved as following:

$$
\begin{aligned}
c_1 &= \frac{\int_\Omega I(x) H_\varepsilon(\phi_k(x))\mathrm{d}x}{\int_\Omega H_\varepsilon(\phi_k(x))\mathrm{d}x}, \\
c_2 &= \frac{\int_\Omega I(x)(1 - H_\varepsilon(\phi_k(x)))\mathrm{d}x}{\int_\Omega (1 - H_\varepsilon(\phi_k(x)))\mathrm{d}x}, \\
f_1(x) &= \frac{K_\sigma(x) * [H_\varepsilon(\phi_k(x)) I(x)]}{K_\sigma(x) * H_\varepsilon(\phi_k(x))}, \\
f_2(x) &= \frac{K_\sigma(x) * [(1 - H_\varepsilon(\phi_k(x))) I(x)]}{K_\sigma(x) * [1 - H_\varepsilon(\phi_k(x))]}
\end{aligned}
\tag{2}
$$

$\phi_k, k = 0, 1, 2\ldots$ denotes the level set function at iteration $k$.

Fixing $c_1, c_2, f_1, f_2$ and using the calculus of variation method, one can have:

$$\nabla_\phi F(\phi_k) = \delta_\varepsilon(\phi_k)\left[\eta_1(u-c_1)^2 - \eta_2(u-c_2)^2 - \nabla \cdot \left(g\frac{\nabla \phi_k}{|\nabla \phi_k|}\right)\right]$$
$$+\lambda_1 \int K_\sigma(x-y)|I(y)-f_1(x)|^2 \delta(\phi_k(y))\mathrm{d}y$$
$$-\lambda_2 \int K_\sigma(x-y)|I(y)-f_2(x)|^2 \delta(\phi_k(y))\mathrm{d}y \tag{3}$$

Thus the iterative formula of the gradient descent method has the form:

$$\phi_{k+1} = \phi_k - \alpha_k \nabla_\phi F(\phi_k) \tag{4}$$

where $\alpha_k$ is the time step length. $d_k := -\nabla_\phi F(\phi_k)$ is the gradient descent direction. Meeting the conditions of convergence, the optimal values of $\phi$ to minimize the energy function (i) is the boundary of the image. Detailed coding information to execute the ACM algorithm is provided in the supplementary software file (Supplementary Note 1), which is implemented in MATLAB R2012a under the Windows XP system.

**Statistics.** For feather follicle manipulation in vivo, at least five follicles were used for each experimental condition and representative results were shown. Data are expressed as mean±s.e.m. The statistical difference between two groups was determined by the two-tailed $t$-test, and the $p$-value was calculated.

**Data availability.** The authors declare that all data supporting the findings of this study are available within the article and its supplementary information files or from the corresponding author upon reasonable request. The RNA-seq data generated in the present study have been deposited in the GEO database under accession code GSE110591.

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

## Acknowledgements

We thank Dr Olivier Pourquie (IGBMC, France) for providing the Notch1 expression plasmid; Dr Fernando Giraldez (Universitat Pompeu Fabra, Spain) for providing the Serrate1 and Serrate2 expression plasmids; Dr Wei Wu (Tsinghua University, China) for providing the Notch reporter plasmid pGa981-6; Dr Hongrui Wang (Xiamen University,

China) for providing the DN and CA forms of Rho/Rac/Cdc42 constructs, Dr Xinhua Liao (Fujian Medical University, China) for technical assistance, and Qingxiang Gao for bioinformatics analysis. This work is supported by start-up funds from Fuzhou University, grants from the National Science Foundation of China (31071285 and 31371274) and Department of Science and Technology of Fujian Province (2017J01630) to Z.Y.

## Author contributions

Z.Y., R.B.W., and C.M.C. conceived the work. D.C., X.Y., G.Q., and J.Z. contributed equally to this work and performed most of the experiments. D.C. characterized the filopodia and analyzed the role of Rho GTPases. X.Y. and J.Z. performed the in situ hybridization analysis. X.Y., G.Q., and T.F. analyzed the role of Notch signaling and E-Cadherin. G.Q. and Y.T. analyzed the role of FGF signaling. J.Z. analyzed the RNAi knockdown efficiency. H.W. and W.H. performed the TEM analysis. H.X. and M.W. analyzed the image. P.W. analyzed the role of Delta. J.Z. helped draft the manuscript. Z.Y., R.B.W., and C.M.C. drafted the manuscript. All authors read and approved the final manuscript.

## Additional information

**Competing interests:** The authors declare no competing interests.

