## [Peer Review File · Nature Communications]

Reviewers' comments:

Reviewer #1 (Remarks to the Author):

The authors have studied the mechanisms underlying the patterning of avian feather development. They show that the initially flat epithelial sheath at the base of the follicle develops periodic branches. The basal keratinocytes of this epithelium have numerous, unusually long filopodia that project into the underlying mesenchyme. These filopodia are regulated by the small GTPases, RhoA and Cdc42, and expressing dominant negative mutants of these GTPases altered both the filopodia and subsequent feather patterning. The authors suggest that the organization of feathers into branches is due to differential cell adhesion, based on the patterning of E-cadherin and beta catenin, as well as the effects of over-expression and knockdown of E-cadherin. They go on to examine the Notch pathway, including the ligands Serrate1 and 2; concluding that this pathway may be involved, perhaps by the usual lateral inhibition mechanism. FGF10 was also shown to play a role. They propose a model in which a proximal-distal gradient of FGF cooperates with lateral inhibition from Notch. Notch is activated only when FGF levels fall below a threshold. The periodically activated Notch drives contraction of filopodia and differential adhesion, which in turn promotes feather branching.

Although this paper could be potentially interesting, the data fall short of supporting the authors' interpretations and conclusions.

1. Dominant negative mutants of small GTPases, such as RhoA or Cdc42, work by binding to and sequestering the guanine nucleotide exchange factors (GEFs) for these GTPases. It has been known for a some years now that these dominant negative mutants cause non-specific effects by binding to and sequestering the GEFs for other, closely related small GTPases. (This reviewer has made this exact mistake long ago, before this artifact was appreciated.) The acceptable experiment is to knock down or knock out the specific small GTPase, with, of course, controls for specificity, etc.

2. The authors use the patterning of E-Cadherin and beta-catenin, and the effects of over-expressing and knocking down E-Cadherin, to conclude that differential adhesion in the epithelium is involved in patterning of feather branching. Adhesion is not measured. Adhesion depends on much more than just the levels of E-Cadherin as determined and manipulated by the methods used here. The supra-molecular organization of E-Cadherin (which is incompletely understood) and its interaction with its binding partners determine adhesion, which is a highly dynamic process.

3. Figure 2c and d. What do the line scans below the micrographs show, and what conclusion is drawn from these scans?

4. Overall model and conclusion. How do the changes in filopodia and border length lead to the shape change shown in Figure 4d'? It is not clear and convincing how a change in adhesion can lead to this shape change. Moreover, it is not clear and convincing how this ultimately results in the changes in feather branch patterning.

5. Minor point. Feather patterning is often said to be a classic example of a Turing reaction-diffusion or similar mechanism for periodic pattern formation. This might be mentioned here, at least in half a sentence or so, to provide some context.

6. The paper is not well presented and is difficult to follow. It is not a matter of the English language, which has only minor mistakes that could be easily fixed. Rather, ideas and data are not presented in a logical order, and are not explained in sequence. There are many gaps in the flow of information. References to figures jump back and forth.

--

Reviewer #2 (Remarks to the Author):

The investigators find that feather branching is dependent on filopodia which are present on early feather structures. As the feather develops the filopodia are lost. The filopodia are controlled by GTPase Rho/Cdc42. The investigators go on to show that the notch signaling pathway and the FGF signaling pathway are important for filopodia formation and feather branching. The investigators use elegant techniques including the injection of lentivirus to create transgenic feathers. To the best of my knowledge, the description of the filopodia in the primitive feathers is new. Similarly, the finding that notch and FGF signaling are important for feather branching and filopodia is relatively new, though perhaps not that surprising since these factors have been studied in other epithelia and shown to be important for appendage formation. Nonetheless, the careful nature of the studies provides solid evidence that these pathways are also important for feather formation.

I do not have any criticisms of the data presented in the manuscript.

--

Reviewer #3 (Remarks to the Author):

In this paper, Cheng et al studied the molecular control of the periodic branching of feathers. First, a morphological analysis of the feather epithelium along its proximo-distal axis defined that cells present in the "pre-branching" epithelium displays extensive filopodia, which are absent in the branching epithelium. They show that filopodia are important for branching and eventually for feather patterning, since dominant-negative forms of the Rho GTPases RhoA and Cdc42 (but not Rac1), known to inhibit filopodia formation, modified this patterning.

Second, they demonstrate that the periodic expression of E-cadherin in the branching epithelium is important for feather patterning.

Third, they present evidence indicating that the periodic expression of Notch family members regulates this branching, possibly by regulating the periodic expression of E-cadherin and beta catenin in the branching epithelium through a lateral inhibition mechanism.

Finally, they show that FGF signaling may play a role in the same process, presumably by regulating filipodia formation.

The paper is interesting, as it presents novel hypotheses on the molecular control of branching during feather formation.

The general feeling while reading the manuscript is that its storyline should be significantly reinforced, and stronger connection between the chapters should be made. As it is, it reads as two separate stories, one on filopodia and FGF (a tiny paragraph on FGF that seems lost in the paper), the second on Notch and cell adhesion. The last chapter on the "coastline paradox" seems also incongruous.

More importantly, there are also serious concerns about the interpretation of the data, in particular on Notch signaling.

Major concerns

"Notch1 and Serrate1 are enriched in the barb plate (...) GFP (Notch reporter) is only expressed in the marginal plate (Fig. 3l-m), indicating specific activation of Notch signaling in this region"

From the text and figures, it seems that Notch1 is expressed in the barb plate (BP), while the reporter and the downstream targets L-Fringe and Hey are expressed in the marginal plate (MP). It does not really make a lot of sense. Could it be that another receptor of the Notch family expressed in the MP accounts for the activation of the Notch targets and reporter in this domain? It would be important to discover which one. Another point that would derive from this is whether the ligands (Serr1 & 2) are acting in trans or cis on the receptor(s). This is significant, since it is known that binding in cis is inhibitory (reviewed in Del Alamo et al. Curr. Biol. 2011) while binding in trans is activatory. Obviously this could significantly change the conclusions from the paper.

The experiments show that Notch signaling regulates periodic branching, but the evidence that this is done through lateral inhibition (lines 118-140) is weak at best. Notch can act on a number of targets and not necessarily through a lateral inhibition mechanism.

The authors should test whether there is a hierarchy in the signals and pathways they uncovered. Is FGF upstream of Notch, downstream, do they act in parallel?

Co-iP and Western blot data: no indication in the text on how those experiments were done, on cells or tissues, is it over-expression? In which case one can wonder on whether the same data (the physical association of Notch and beta-catenin; the down-regulation of E-Cadherin and beta catenin) would be obtained in an in vivo environment. This should be verified.

The efficiency of siRNAs was tested in DF1 chicken fibroblasts. The authors should verify in an in vivo environment (by in situ hybridization) that the endogenous transcript is absent.

The authors should find a way to show the filopodia in a much clearer way. As it is, it is hard for non-specialists to understand what they show. Maybe they could infect the follicle with lentiviruses expressing a membrane bound GFP, or a LifeAct GFP.

Other concerns

Figure 3L: one assumes that PLL3.7 staining identifies all the cells that were infected, but it does not seem to be explained how this was done anywhere.

"Higher magnification views showed clear basal lamina along these filopodia, including lamina densa and lamina lucida": Maybe for EM specialists but not for regular biologists.

"On average, each basal cell extends 3~5 filopodia about 2~10 μm long". How was this quantified?

"We demonstrated sSer2 can reduce the Notch reporter activity in cell culture". It's been long shown that secreted Delta and Serrate ligand inhibit Notch activity, such that this is only a confirmation that the tool is ok.

How are the phenotypes quantified? Does, for instance a 10/17 means that only 17 feather buds were infected and analyzed (where 10 showed the indicated phenotype)? This would be at least an order of magnitude below what is expected at this level of publication.

Obscure to non-specialists, please re-phrase

- "Calculating the surface area before and after branching reveals a scaling effect resembling the "coastline paradox": when we zoom in, more details emerge and increase the complexity of an object. Thus the surface area increase in feather branching morphogenesis is actually prepared in advance". It is doubtful that most biologists know what the coastline paradox of Benoit Mandelbrot is, such that some more explanations would be helpful.

- "Recently, this exquisite structure was harnessed as a means of recording and dissecting the pathological principles in radio- and chemotherapy".

Reviewers' comments:

Reviewer #1 (Remarks to the Author):

The authors have studied the mechanisms underlying the patterning of avian feather development. They show that the initially flat epithelial sheath at the base of the follicle develops periodic branches. The basal keratinocytes of this epithelium have numerous, unusually long filopodia that project into the underlying mesenchyme. These filopodia are regulated by the small GTPases, RhoA and Cdc42, and expressing dominant negative mutants of these GTPases altered both the filopodia and subsequent feather patterning. The authors suggest that the organization of feathers into branches is due to differential cell adhesion, based on the patterning of E-cadherin and beta catenin, as well as the effects of over-expression and knockdown of E-cadherin. They go on to examine the Notch pathway, including the ligands Serrate1 and 2; concluding that this pathway may be involved, perhaps by the usual lateral inhibition mechanism. FGF10 was also shown to play a role. They propose a model in which a proximal-distal gradient of FGF cooperates with lateral inhibition from Notch. Notch is activated only when FGF levels fall below a threshold. The periodically activated Notch drives contraction of filopodia and differential adhesion, which in turn promotes feather branching.

Although this paper could be potentially interesting, the data fall short of supporting the authors' interpretations and conclusions.

1. Dominant negative mutants of small GTPases, such as RhoA or Cdc42, work by binding to and sequestering the guanine nucleotide exchange factors (GEFs) for these GTPases. It has been known for a some years now that these dominant negative mutants cause non-specific effects by binding to and sequestering the GEFs for other, closely related small GTPases. (This reviewer has made this exact mistake long ago, before this artifact was appreciated.) The acceptable experiment is to knock down or knock out the specific small GTPase, with, of course, controls for specificity, etc.

We thank this reviewer for pointing out our oversight on this issue. We now performed RNAi knockdown experiments for each of the three GTPases, with analysis of the specificity and knockdown efficiency, and examined their functional roles in feather branching morphogenesis. The results are organized in the new Fig. 2 and Supplementary Fig. 5.

In summary, the results are consistent with overexpression of the dominant negative forms. RNAi knockdown of RhoA and Cdc42, but not Rac1, disrupt the feather rachis.

In the text, we incorporated these new results (Page 3), and cited the relevant references (Feig, 1999 NCB; Debreceni et al., 2004 JBC).

(Page 5)

"Because the DN forms of GTPases may elicit non-specific effects, we further verified the impact of RNAi knockdown of these small GTPases. The knockdown

efficiency and specificity of RNAi were each verified in vitro (Supplementary Fig. 5). Consistent with the results from overexpressing the DN forms, knockdown of RhoA or Cdc42 produced feathers with weaker or no rachis in the upper part, whereas knockdown of Rac1 resulted in normal feathers (Fig. 2e)."

2. The authors use the patterning of E-Cadherin and beta-catenin, and the effects of over-expressing and knocking down E-Cadherin, to conclude that differential adhesion in the epithelium is involved in patterning of feather branching. Adhesion is not measured. Adhesion depends on much more than just the levels of E-Cadherin as determined and manipulated by the methods used here. The supra-molecular organization of E-Cadherin (which is incompletely understood) and its interaction with its binding partners determine adhesion, which is a highly dynamic process.

Again, thanks for this very good point that E-Cadherin expression levels are not equal to the real adhesion force. Since it is not practical at this moment to actually measure the adhesion forces in feather development, we analyzed the cell adhesion changes and the organization of E-Cadherin molecules in more detail.

It turns out that by TEM analysis, E-Cadherin molecules between two adjacent basal keratinocytes do form AJs before branching, either in the lateral surface below the TJs (similar to our classical view of epithelial cell adhesions; Fig. 1d), or between filopodia (Fig. 3f). However, it appears that the AJs in filopodia are more transient and dynamic structures akin to the "nanoclusters" (Yap et al., 2015). In the branched feather barb, the outer layer cells (marginal plate) uses more stable TJs and desmosomes to build cell connections, as these cells have reduced levels of E-Cadherin (Supplementary Fig. 2b). The inner barbs cells, which have higher levels of E-Cadherin, showed distinct punta of E-Cadherin distribution (Fig. 3e), and continuous zones of AJs under TEM (Fig. 3f). These structures resemble the so-called "microclusters" (Yap et al., 2015) or the adhesion zipper structure (Vasioukhin et al., 2000), as described previously.

We incorporated these new data and concepts with re-organized figures (Fig. 1d; Fig. 3e,f; Supplementary Fig. 2) and modified the text (Page 5-6):

"Because E-Cadherin-mediated cell adhesion depends on its organization at the nanoscale, we examined in detail its distribution pre- and post- feather branching. In the pre-branch basal keratinocytes, the E-Cadherin molecules between two adjacent cells do form AJs; however, it appears these AJs are more transient and dynamic structures akin to the "nanoclusters" (Fig. 3e,f). On the other hand, in the branched feather barbs, the outer layer cells (marginal plate) uses more stable TJs and desmosomes to build cell connections, as these cells have reduced levels of E-Cadherin (Supplementary Fig. 2b). The inner barb cells, which have higher levels of E-Cadherin, showed distinct punta of E-Cadherin distribution (Fig. 3e), and continuous zone of AJs under TEM (Fig. 3f). These structures resemble the so-called "microclusters" or the previously described adhesion zipper structure. Thus there are both E-Cadherin down-regulation in

the basal keratinocytes and re-distribution in the suprabasal cells during feather branching.”

3. Figure 2c and d. What do the line scans below the micrographs show, and what conclusion is drawn from these scans?

We removed these features from the figures. The initial intention is to show the re-distribution of E-Cadherin and β -Catenin in the branching process. However, these points are self-evident in the figures (Fig. 3c,d).

4. Overall model and conclusion. How do the changes in filopodia and border length lead to the shape change shown in Figure 4d? It is not clear and convincing how a change in adhesion can lead to this shape change. Moreover, it is not clear and convincing how this ultimately results in the changes in feather branch patterning.

To better illustrate our concept, we modified the model figures (Fig. 5f), with new data to show that the filopodia are involved in sense/transport FGF10 molecules (Fig. 5c,d). Thus the filopodia in basal keratinocytes not only prepare the surface area for branching morphogenesis, but also form a positive feedback loop to reinforce the FGF signaling gradient in the proximal follicle.

In summary, there are two related but largely independent cellular aspects that will impact feather branching: the filopodia, and E-Cadherin-mediated cell adhesion (as shown in Fig. 2 and Fig. 3). These two aspects are in turn, regulated by two related but largely independent signaling pathways, Notch and FGF signaling (as shown in Fig. 4 and Fig. 5). In Fig. 4n, we have shown that the activation of Notch signaling leads to the contraction of filopodia and reduced E-Cadherin expression in the marginal plate cells. In Fig. 5, we have shown that the filopodia is involved in transporting FGF molecules, and Notch activation only occurs below a FGF signaling threshold.

In light of the these data, it appears that the basal filopodia not only prepare the surface area for branch formation, but also represent a competent cell status where the cells actively explore the microenvironment, possibly to sense the FGF gradient. On the other hand, Notch activation drives filopodia contraction and E-Cadherin down-regulation/re-distribution for branch formation. Thus under a classical “activator-inhibitor” view, FGF signaling may serve as the “inhibitor” to block feather branching, whereas Notch activation serves as the “activator” to drive branch formation. Contraction of the filopodia may facilitate the activation of Notch signaling and feather branching via reduced FGF signaling.

We have incorporated these concepts and considerations into the revised Discussion, which we hope will clarify the topic (Page 10-12).

5. Minor point. Feather patterning is often said to be a classic example of a Turing reaction-diffusion or similar mechanism for periodic pattern formation. This might be mentioned here, at least in half a sentence or so, to provide some context.

Thank you for this point. We have elaborated this concept in both the Introduction (Page 3) and Discussion (Page 10-12).

6. The paper is not well presented and is difficult to follow. It is not a matter of the English language, which has only minor mistakes that could be easily fixed. Rather, ideas and data are not presented in a logical order, and are not explained in sequence. There are many gaps in the flow of information. References to figures jump back and forth.

We have made significant changes to both the Figures and Text to smooth out the logic flow.

--

Reviewer #2 (Remarks to the Author):

The investigators find that feather branching is dependent on filopodia which are present on early feather structures. As the feather develops the filopodia are lost. The filopodia are controlled by GTPase Rho/Cdc42. The investigators go on to show that the notch signaling pathway and the FGF signaling pathway are important for filopodia formation and feather branching. The investigators use elegant techniques including the injection of lentivirus to create transgenic feathers. To the best of my knowledge, the description of the filopodia in the primitive feathers is new. Similarly, the finding that notch and FGF signaling are important for feather branching and filopodia is relatively new, though perhaps not that surprising since these factors have been studied in other epithelia and shown to be important for appendage formation. Nonetheless, the careful nature of the studies provides solid evidence that these pathways are also important for feather formation.

I do not have any criticisms of the data presented in the manuscript.

Thank you very much for appreciating our work.

--

Reviewer #3 (Remarks to the Author):

In this paper, Cheng et al studied the molecular control of the periodic branching of feathers. First, a morphological analysis of the feather epithelium along its proximo-distal axis defined that cells present in the "pre-branching" epithelium displays extensive filopodia, which are absent in the branching epithelium. They show that filopodia are important for branching and eventually for feather patterning, since dominant-negative forms of the Rho GTPases RhoA and Cdc42 (but not Rac1), known to inhibit filopodia formation, modified this patterning. Second, they demonstrate that the periodic expression of E-cadherin in the branching epithelium is important for feather patterning. Third, they present evidence indicating that the periodic expression of Notch family members regulates this branching, possibly by regulating the periodic expression of E-cadherin and beta catenin in the branching epithelium through a lateral inhibition mechanism. Finally, they show that FGF signaling may play a role in the same process, presumably by regulating filipodia formation.

The paper is interesting, as it presents novel hypotheses on the molecular control of branching during feather formation.

The general feeling while reading the manuscript is that its storyline should be significantly reinforced, and stronger connection between the chapters should be made. As it is, it reads as two separate stories, one on filopodia and FGF (a tiny paragraph on FGF that seems lost in the paper), the second on Notch and cell adhesion. The last chapter on the "coastline paradox" seems also incongruous.

Thank you very much for appreciating our work. We have made significant changes to both the Figures and Text to smooth out the storyline.

More importantly, there are also serious concerns about the interpretation of the data, in particular on Notch signaling.

Major concerns

"Notch1 and Serrate1 are enriched in the barb plate (...) GFP (Notch reporter) is only expressed in the marginal plate (Fig. 3l-m), indicating specific activation of Notch signaling in this region"

From the text and figures, it seems that Notch1 is expressed in the barb plate (BP), while the reporter and the downstream targets L-Fringe and Hey are expressed in the marginal plate (MP). It does not really make a lot of sense. Could it be that another receptor of the Notch family expressed in the MP accounts for the activation of the Notch targets and reporter in this domain? It would be important to discover which one. Another point that would derive from this is whether the ligands (Serr1 & 2) are acting in trans or cis on the receptor(s). This is significant, since it is known that binding in cis is inhibitory (reviewed in Del Alamo et al. Curr. Biol. 2011) while binding in trans is activatory.

Obviously this could significantly change the conclusions from the paper.

Thank you for pointing this out. Actually we are aware of this potential pitfall in our Manuscript, and that is why we did extensive characterization of Notch activation in the feather follicle (Figure 4). It turns out that our initial RNA-seq experiment through BGI (Beijing Gene Institute, ShengZhen, China) was not ideal: the coverage rate of genes is about 89.5%, which missed out Notch2. We have thus performed a second round of RNA-seq experiment through another vendor, Novogene (Beijing, China). This time Notch2 was shown to be expressed, at comparable overall levels to Notch1, but with a broader distribution pattern (Fig. 4b; Supplementary Table 1). Importantly, Notch2 is expressed in the basal keratinocytes as well as the marginal plate cells, thus may serve as the receptors for Notch activation (which is specifically in the marginal plate cells, as shown by both downstream gene expression, and a GFP-reporter assay in vivo; Fig. 4i,j,k,l). We further evaluated the functional significance of Notch2 in vivo (Fig. 4g and Supplementary Fig. 8).

With these new data, we re-organized the Figures (Fig. 4, Supplementary Fig. 7, Supplementary Fig. 8, Supplementary Table 1), and re-write the Discussion with new references (Page 10-12):

“Given the complex expression patterns of the various Notch ligands and receptors in the feather follicle, and the potential cis- and trans- interactions of the ligands/receptors, the activation of Notch signaling in the feather follicle is likely to be complicated. We have examined the impact of overexpression and knock-down of both the receptors (Notch1, Notch2) and ligands (Ser1, Ser2). It appears Notch2 is the endogenous receptor that is responsible for the activation of Notch signaling in the marginal plate keratinocytes, Ser1 serves as the ligand to drive its activation, whereas Ser2 acts in cis to inhibit its activation. The expression of L-Fringe may further modulate Notch activation. Our results are consistent with the current understanding of Notch signaling activation: overexpression of Notch receptors inhibited branching, because the limited amount of endogenous ligands (Ser1, Ser2) will be sequestered in cis, thus reduced trans-activation of Notch signaling; conversely, knockdown of Notch receptors will lead to more available ligands for trans-activation of Notch signaling. Similarly, because Ser1/2 is inhibitory in cis for Notch activation, knockdown of these molecules will promote Notch activation and branch formation.”

The experiments show that Notch signaling regulates periodic branching, but the evidence that this is done through lateral inhibition (lines 118-140) is weak at best. Notch can act on a number of targets and not necessarily through a lateral inhibition mechanism.

See Discussion above. We have modified our interpretation in the Manuscript, and discussed in detail how Notch signaling is activated to drive feather branching.

The authors should test whether there is a hierarchy in the signals and pathways they uncovered. Is FGF upstream of Notch, downstream, do they act in parallel?

We tested the relationship between FGF and Notch activation both in vivo and in vitro. These two signaling pathways are largely independent but related. For instance, high FGF level blocks feather branching (thus Notch activation; Fig. 5b); consistently, FGF reduces Notch reporter activity in 293T cells (Supplementary Fig. 11d). On the other hand, Notch activation may negatively feedback on FGF signaling, with the reduced basal filopodia (Fig. 4n), but only under a permissive FGF threshold. Contraction of the filopodia may facilitate the activation of Notch signaling and feather branching via reduced FGF signaling. These considerations are summarized in the new model (Fig. 5f), and the revised Discussion (Page 10-12).

Co-iP and Western blot data: no indication in the text on how those experiments were done, on cells or tissues, is it over-expression? In which case one can wonder on whether the same data (the physical association of Notch and beta-catenin; the down-regulation of E-Cadherin and beta catenin) would be obtained in an in vivo environment. This should be verified.

We have moved these in vitro data (in cell culture) to Supplementary Fig. 11, with detailed description of the experiments.

We also incorporated new in vivo data to show that ectopic Notch activation leads to contraction of filopodia and down-regulation of E-Cadherin (Fig. 4n).

The efficiency of siRNAs was tested in DF1 chicken fibroblasts. The authors should verify in an in vivo environment (by in situ hybridization) that the endogenous transcript is absent.

We now performed this experiment and the data were shown in Supplementary Fig. 9. The data showed efficient knockdown of the specific genes in the feather follicle in vivo.

The authors should find a way to show the filopodia in a much clearer way. As it is, it is hard for non-specialists to understand what they show. Maybe they could infect the follicle with lentiviruses expressing a membrane bound GFP, or a LifeAct GFP.

The filopodia in basal keratinocytes is an in vivo phenomenon in the pre-branch feather epithelium. So far in our hands, in vitro culture of feather branching is not successful (an effort first carried out by Lillie & Wang in the 1940s but was unsuccessful and abandoned; thus live-imaging of the filopodia during feather branching is not practical, at least for now). Thus we can only show this by sectioning the samples. In this situation, we surmise that even with LifeAct GFP indication, we still need to cut sections for the sample to show these features, which is not fundamentally different from what we have done here (i.e. TEM, staining, etc.).

Other concerns

Figure 3L: one assumes that PLL3.7 staining identifies all the cells that were infected, but it does not seem to be explained how this was done anywhere.

We have incorporated explanation both in the Text and in the Figure legend:
(Page 7 Line 170)

"For control, a viral vector where GFP expression was driven by a CMV promoter showed widespread virus expression."

(Page 24 Line 544)

"Expression of a 6XCSL-GFP reporter was specifically in the marginal plate of branching feather epithelium, whereas a control viral vector showed widespread virus expression in the follicle."

"Higher magnification views showed clear basal lamina along these filopodia, including lamina densa and lamina lucida": Maybe for EM specialists but not for regular biologists.

In Fig.1c, we now marked these two layers of the basal lamina and explained in the legend (Page 21 Line 503):

"Basal lamina were shown in higher magnifications in c, with the lamina lucida indicated by dots, and lamina densa indicated by arrow heads."

"On average, each basal cell extends 3~5 filopodia about 2~10 μ m long". How was this quantified?

These were quantified through analyze the TEM images of the basal keratinocytes, which were verified with the regular staining results. This is intended to distinguish the filopodia from the notion of a single, long projection such as the cilia structure or neurites. On Page 4 Line 71:

"On average, each basal cell extends 3~5 filopodia about 2~10 μ m long as counted/measured from the TEM images, with no single filopodium in dominance of the others."

"We demonstrated sSer2 can reduce the Notch reporter activity in cell culture". It's been long shown that secreted Delta and Serrate ligand inhibit Notch activity, such that this is only a confirmation that the tool is ok.

We moved these data to Supplementary Fig. 10 as supporting information, and modified Fig. 4 accordingly.

How are the phenotypes quantified? Does, for instance a 10/17 means that only 17 feather buds were infected and analyzed (where 10 showed the indicated phenotype)? This would be at least an order of magnitude below what is expected at this level of publication.

For 10/17, we infected 17 feather follicles in three-months old chicken with lentiviral injection, and 10 showed the indicated phenotype. Since the method we used is not through genetic manipulation of the chicken, but via ectopic viral transduction in the adult animal, the efficiency of our manipulation is already very high as compare to other virus infection methods, such as RCAS used by previous work (Yu et al., Nature 2002: ref 9; Yue et al., PNAS 2006: ref 17; Yue et al., Dev Biol 2012: ref 18; Li et al., Nature Comm 2017: ref 6). Indeed, the incidence of phenotype after Notch signaling pathway perturbation (about 50% of the cases) is higher than Wnt/Dkk pathway perturbation using the same method (Chu et al., Dev Biol 2014: ref 19; about 10-20% of the cases).

Obscure to non-specialists, please re-phrase

- *"Calculating the surface area before and after branching reveals a scaling effect resembling the "coastline paradox": when we zoom in, more details emerge and increase the complexity of an object. Thus the surface area increase in feather branching morphogenesis is actually prepared in advance". It is doubtful that most biologists know what the coastline paradox of Benoit Mandelbrot is, such that some more explanations would be helpful.*

We explained in more detail in the Abstract and Result.

(Page 1)

"Calculating the surface area before and after branching reveals a scaling effect resembling the "coastline paradox", which was proposed by Benoit Mandelbrot in the 1960s to describe the fractal nature of the coastline."

(Page 9)

"This situation resembles the coastline paradox, which claims that a given landmass may not have a fixed coastline length because of the fractal-like property of its coastline."

- *"Recently, this exquisite structure was harnessed as a means of recording and dissecting the pathological principles in radio- and chemotherapy".*

We re-phrased this introductory sentence, and cited relevant references (Page 3):
“Recently, the regularly branched feather structure was utilized as a model to dissect the pathological principles of tissue damage in chemo- and radiation therapy, because any perturbation of feather development will be recorded in its final morphology.”

Reviewers' comments:

Reviewer #1 (Remarks to the Author):

I am sorry to say that significant problems remain with the data in this paper.

RNAi for Rho family GTPases.

Fig. S5. The controls were performed on a fibroblast line, which is irrelevant to the in vivo experiments. Efficiency for Cdc42 was less than 40%, and so it would be surprising if there would be any relevant effect.

There was no attempt to assess efficiency of knockdown in vivo and so the experiment cannot be assessed.

There are about 20 Rho family GTPases. DN and CA mutants can act promiscuously, as can RNAi, if not properly controlled for. There was no indication of proper controls here.

E-Cadherin.

Fig. 3. "Nanoclusters" are allegedly seen in some panels. As far as this author is aware, nanoclusters of E-cadherin are seen with super-resolution light microscopy, which was not used in this paper.

RNAi knockdown of E-cadherin was also quantitated in a fibroblast cell line (Fig. S5). Fibroblasts tend to express relatively low levels of E-Cadherin. In contrast, epithelial cells have high levels of E-Cadherin. In this reviewer's experience, knockdown or otherwise inactivating E-cadherin in epithelial cells in vivo is extremely difficult. As there does not appear to be any assessment of the degree of knockdown or inactivation of E-cadherin in vivo, this experiment cannot be assessed.

--

Reviewer #3 (Remarks to the Author):

The authors have made extensive changes and performed many experiments to address the numerous comments of the reviewers. As a result, the quality of the study is significantly increased. I do not have any additional comment on the data at this point. Well done!...

Reviewers' comments:

Reviewer #1 (Remarks to the Author):

I am sorry to say that significant problems remain with the data in this paper.

RNAi for Rho family GTPases.

Fig. S5. The controls were performed on a fibroblast line, which is irrelevant to the in vivo experiments. Efficiency for Cdc42 was less than 40%, and so it would be surprising if there would be any relevant effect.

There was no attempt to assess efficiency of knockdown in vivo and so the experiment cannot be assessed.

There are about 20 Rho family GTPases. DN and CA mutants can act promiscuously, as can RNAi, if not properly controlled for. There was no indication of proper controls here.

Thank you very much for the helpful comments. We have now carefully addressed this issue which we believe have significantly improved our feather model system from a technical perspective.

(1) We used both in situ hybridization and quantitative RT-PCR to evaluate the RNAi knockdown efficiency in vivo in the feather follicle. It turns out the in vivo knockdown efficiency is comparable to the results obtained from the DF-1 chicken cell line, possibly because the virus infection is very efficient (although variations between follicles were more significant than in cell cultures). The results are now summarized in Supplemental Fig. S5 & Fig. S6 (the in vivo knockdown efficiency for Notch related molecules were also examined and results presented in Fig. S5).

In the Methods section, we stated (Page 14):

To examine the knockdown efficiency, the constructs were electroporated into DF-1 cells and total RNAs were extracted 48 hours later. For E-Cadherin, the full length chicken cDNA was cloned into the pEGFP-N1 expression plasmid and co-electroporated with the RNAi construct. To examine the knockdown efficiency in vivo, virus infection was performed in plucked feather follicles and samples were collected 4 days post-infection. Each follicle was individually collected and total RNAs extracted for qRT-PCR analysis.

(2) To address the specificity of RNAi knockdown, we first examined through RNA-seq the gene expression levels of the known Rho GTPases in the feather follicle. For the over 20 Rho family GTPases, 10 were expressed at significant levels in the developing feather follicle (see Supplementary Table 1). We then examined the impact of RNAi on all these GTPases, which confirmed the specificity of RNAi both in vitro (DF-1 cells) and in vivo (the feather follicle). The results are now summarized in Supplemental Fig. S6). We also cited relevant references (Haga RB, Ridley AJ. Rho GTPases: Regulation and roles in cancer cell biology. *Small GTPases*. 7(4): 207-221, 2016.; Ridley AJ. Rho GTPase signalling in cell migration. *Curr Opin Cell Biol*. 36:103-112, 2015).

In the Results section, we stated (Page 5):

Because the DN forms of GTPases may elicit non-specific effects (20, 21), and there are over 20 Rho family GTPases in the avian genome (Supplementary Table 1) (22, 23), we further verified the impact of RNAi knockdown of these small GTPases. The knockdown efficiency and specificity of RNAi were each verified in vitro and in vivo (Supplementary Fig.

5, Fig. 6). In particular, RNAi for RhoA or Cdc42 did not perturb the expression of other Rho family GTPases (Supplementary Fig. 6).

(3) To improve the knockdown efficiency for Cdc42 and Rac1, we have now examined at least four different sites for each of the three GTPases. The best knockdown efficiency is now about 60%, both in vitro and in vivo (Fig. S5; Fig. S6). The new target sites are now listed in Materials & Methods. We obtained similar phenotypes from previous work in vivo (the representative feathers showing the phenotypes in Fig. 2 are now replaced with results from the new virus constructs). Retrospectively, the reason we can see phenotypes even with lower knockdown efficiency could be because the three GTPases, in particular RhoA and Cdc42 are expressed at high levels in the developing feather follicle (Supplementary Table 1), and the filopodia is sensitive to the levels of these GTPases. On the other hand, we have some virus constructs that can knockdown gene expression highly efficiently yet produced normal feather without obvious phenotype (e.g. Dkk3; ref 19). Therefore, it is not possible at this moment to predict which gene or construct will disrupt normal feather development in vivo.

The new target sites are now described in the Methods section (Page 14).

(4) For control RNAi, we used a construct target a random sequence which did not perturb feather development (Chu et al., Dev Biol 2014; ref 19). This type of control RNAi construct is acceptable in current publications (e.g. Guo W et al., Nat Commun 8(1): 2168, 2017; Vilaboa N et al., Nucleic Acids Res 45: 5797, 2017.). This is now stated in the Methods (Page 14):

A scramble control (agatcagcagaggacact) was used (ref 19).

With these additional works, we hope you find the RNAi experiments are now well-defined and properly controlled, and thus the data and analysis are justified.

E-Cadherin.

Fig. 3. "Nanoclusters" are allegedly seen in some panels. As far as this author is aware, nanoclusters of E-cadherin are seen with super-resolution light microscopy, which was not used in this paper.

RNAi knockdown of E-cadherin was also quantitated in a fibroblast cell line (Fig. S5). Fibroblasts tend to express relatively low levels of E-Cadherin. In contrast, epithelial cells have high levels of E-Cadherin. In this reviewer's experience, knockdown or otherwise inactivating E-cadherin in epithelial cells in vivo is extremely difficult. As there does not appear to be any assessment of the degree of knockdown or inactivation of E-cadherin in vivo, this experiment cannot be assessed.

We have now re-phrased our Manuscript and avoid claiming "nanoclusters" in our results (Page 6):

In the pre-branch basal keratinocytes, AJs were formed between two adjacent cells; however, it appears the E-Cadherin molecules were more diffusively distributed (Fig. 3e,f).

Page 24 (Fig. 3 legend):

Higher magnification views of regions in (d) showing E-Cadherin was diffusively distributed in pre-branch feather epithelium, but as puncta in branched barbs (e).

For E-Cadherin knockdown efficiency, we first performed the quantification in DF-1 cells (correctly, a fibroblast cell line). We cloned the full-length chicken E-Cadherin gene under a CMV promoter and co-electroporated into DF-1 cells with the RNAi construct (thus not the endogenous E-Cadherin was quantified). We (Xie et al., JID 2015; ref 16) and others (Kim et al., Nat Biotech 23: 222, 2005; Miri et al., Development 140: 4480, 2013) have used similar methods before. We stated in the Methods (Page 14):

For E-Cadherin, the full length chicken cDNA was cloned into the pEGFP-N1 expression plasmid and co-electroporated with the RNAi construct.

With our advancement of methods to measure RNAi knockdown efficiency in vivo, we now have examined the levels of E-Cadherin in the developing feather follicles via antibody staining (because it is eventually the level of proteins that matters) and qRT-PCR. The results were now summarized in Supplemental Fig. S5. The knockdown efficiency is about 70% (qRT-PCR via in vivo specimens), which is achieved after testing multiple different sites in the gene.

In our previous work, we have used blocking peptide and blocking antibody to inhibit E-Cadherin function in adult mice (Xie et al., JID 137: 1731, 2017 and references therein). Others have used truncated forms of the molecule (Reintsch et al., J Cell Biol 170: 675, 2005), antisense morpholino oligos (Sonawane et al., Development 136: 1231, 2009), and conditional knockout (Tinkle et al., PNAS 105: 15405, 2008) to manipulate the E-cadherin levels in vivo. RNAi-mediated knockdown of E-cadherin in human cells in vitro and in xenograft tumor models have been quite successful (e.g. Onder et al., Cancer Res 68: 3645, 2008). Given the vital role of E-cadherin in embryonic development, knockout of this gene is embryonic lethal. Thus, transiently knockdown its expression in the feather follicle offers the unique opportunity to investigate its role in morphogenesis in this system.

--

Reviewer #3 (Remarks to the Author):

The authors have made extensive changes and performed many experiments to address the numerous comments of the reviewers. As a result, the quality of the study is significantly increased. I do not have any additional comment on the data at this point. Well done!...

Thank you for appreciating our work.

REVIEWERS' COMMENTS:

Reviewer #1 (Remarks to the Author):

The paper is now acceptable for publication.

REVIEWERS' COMMENTS:

Reviewer #1 (Remarks to the Author):

The paper is now acceptable for publication.

Thank you very much for appreciating our work.